# Gradient tungsten-doped Bi$_3$TiNbO$_9$ ferroelectric photocatalysts with additional built-in electric field for efficient overall water splitting

Jie Huang[1,2,7], Yuyang Kang[1,7], Jianan Liu[1,2,7], Tingting Yao[1], Jianhang Qiu[1], Peipei Du[3], Biaohong Huang[1], Weijin Hu[1,2], Yan Liang[1], Tengfeng Xie[4], Chunlin Chen[1], Li-Chang Yin[1], Lianzhou Wang[5], Hui-Ming Cheng[1,6] & Gang Liu[1,2] ✉

Bi$_3$TiNbO$_9$, a layered ferroelectric photocatalyst, exhibits great potential for overall water splitting through efficient intralayer separation of photogenerated carriers motivated by a depolarization field along the in-plane $a$-axis. However, the poor interlayer transport of carriers along the out-of-plane $c$-axis, caused by the significant potential barrier between layers, leads to a high probability of carrier recombination and consequently results in low photocatalytic activity. Here, we have developed an efficient photocatalyst consisting of Bi$_3$TiNbO$_9$ nanosheets with a gradient tungsten (W) doping along the $c$-axis. This results in the generation of an additional electric field along the $c$-axis and simultaneously enhances the magnitude of depolarization field within the layers along the $a$-axis due to strengthened structural distortion. The combination of the built-in field along the $c$-axis and polarization along the $a$-axis can effectively facilitate the anisotropic migration of photogenerated electrons and holes to the basal {001} surface and lateral {110} surface of the nanosheets, respectively, enabling desirable spatial separation of carriers. Hence, the W-doped Bi$_3$TiNbO$_9$ ferroelectric photocatalyst with Rh/Cr$_2$O$_3$ cocatalyst achieves an efficient and durable overall water splitting feature, thereby providing an effective pathway for designing excellent layered ferroelectric photocatalysts.

Converting solar energy to hydrogen by means of photocatalysis is one of the promising ways to solve energy and environmental issues[1–5]. Although numerous photocatalysts have been searched and designed for overall water splitting in the past decades, the conversion efficiency of solar energy to hydrogen is still low due to the strong recombination of photogenerated carriers. Recently, ferroelectric materials (e.g., PbTiO$_3$, BiFeO$_3$, Na$_{0.5}$Bi$_{0.5}$TiO$_3$, and Bi$_3$TiNbO$_9$) with built-in electric field that could facilitate photogenerated carrier separation have

attracted wide attention in the field of photocatalysis[6–9]. Among them, Bi$_3$TiNbO$_9$ is an Aurivillius-type layered ferroelectric photocatalyst, which possesses a structural distortion induced depolarization field along the in-plane $a$-axis arising from the dipole interaction between bismuth in the (Bi$_2$O$_2$)$^{2+}$ layer and oxygen in the (BiTiNbO$_7$)$^{2-}$ perovskite layer (Fig. 1a)[10,11]. Previous studies of Bi$_3$TiNbO$_9$ mainly focused on the effect of depolarization field on the photogenerated carrier separation and defect engineering for surface active site modification. These

**Fig. 1 | Crystal structure along *ac* plane. a** Bi$_3$TiNbO$_9$. **b** Bi$_3$TiNbO$_9$-W ($P_s$-ferroelectric polarization, *E*-additional electric field).

synthesized Bi$_3$TiNbO$_9$ based photocatalysts have demonstrated good potential in photocatalytic water reduction or oxidation with the assistance of sacrificial reagents and CO$_2$ reduction[12–14]. In our latest work, Bi$_3$TiNbO$_9$ has revealed efficient and stable photocatalytic overall water splitting by selective exposure of a robust (BiTiNbO$_7$)$^{2-}$ perovskite layer for effective transfer of electron to the cocatalyst, affirming the great promise of Bi$_3$TiNbO$_9$ as a photocatalyst for solar energy to hydrogen conversion[15].

The (Bi$_2$O$_2$)$^{2+}$ and (BiTiNbO$_7$)$^{2-}$ layers in Bi$_3$TiNbO$_9$ stack along the *c*-axis direction, resulting in a tendency to expose the {001} facet as the basal surface and the formation of a nanosheet morphology. The photogenerated carriers exhibit an anisotropic migration property in the Bi$_3$TiNbO$_9$ nanosheet photocatalysts, where the photogenerated electrons usually reach the basal {001} facet across the interlayer, but the photogenerated holes prefer to reach the lateral {110} facet within the intralayer[13,16]. Due to the large interlayer barrier, the photogenerated electrons migrate slowly along the *c*-axis, which increases the recombination probability of the photogenerated carriers[17]. Although reducing the thickness of the layered material along the *c*-axis to obtain an ultrathin structure[18] or selective exposure of different layers can shorten the migration distance of photogenerated electrons from the bulk to surface[15], to some extent, the nature of poor interlayer charge transport remains unchanged. Therefore, the problem of large interlayer charge transport barrier represents an intrinsic bottleneck in designing efficient layered Bi$_3$TiNbO$_9$ photocatalysts for overall water splitting.

Here, we introduced an additional built-in electric field, perpendicular to the depolarization field in Bi$_3$TiNbO$_9$ nanosheets, by gradient tungsten (W) element doping induced energy band structure adjustment between surface and bulk to break such a bottleneck. Typically, donor dopants can effectively increase the number of free electrons, thus raising the corresponding Fermi level of semiconductors[19–23]. The concentration of this donor dopant W$^{6+}$ shows a gradient decrease from surface to bulk along the *c*-axis (Fig. 1b), giving rise to a doping-depth-dependent variation in the gap between the Fermi level and the conduction band minimum (CBM) from surface to bulk and thus a formation of additional built-in electric field along the *c*-axis. This introduced electric field could be used for conquering the potential barrier between the layers of (Bi$_2$O$_2$)$^{2+}$ and (BiTiNbO$_7$)$^{2-}$ for the photogenerated electrons, motivating the photogenerated electrons to the basal {001} facet. Moreover, the W dopant could strengthen the structural distortion, resulting in an enhanced depolarization field along the *a*-axis for the anisotropic flow of the photogenerated carriers. Combined with selective exposure of the robust (BiTiNbO$_7$)$^{2-}$ perovskite layer, W$^{6+}$ doped Bi$_3$TiNbO$_9$ exhibits efficient and stable photocatalytic overall water splitting, which is 10.4 times higher than that of the original Bi$_3$TiNbO$_9$. Our work systematically studied the regulation of the crystal structure and electronic structure arising from element doping in Bi$_3$TiNbO$_9$, providing a reference for the application of Aurivillius compound in photocatalytic overall water splitting.

## Results

### Photocatalysts characterization

W$^{6+}$ ion doped Bi$_3$TiNbO$_9$ samples were prepared by a conventional flux method[24]. Due to the layered structure and the influence of anions in molten salt, all the Bi$_3$TiNbO$_9$-*x*W (*x* = 0, 1%, 3%, 5%, 7%, 12% in molar ratio) samples exhibit a typical nanosheet morphology with lateral width of *ca.* 1 μm and thickness of *ca.* 50 nm (Supplementary Fig. 1), indicating a trivial effect of W doping on the morphology. The inference of the crystal structure was confirmed by the X-ray diffraction (XRD) characterization. As shown in Supplementary Fig. 2a, the diffraction patterns of all samples are basically the same as the standard card (PDF#39-0233, space group: A2$_1$am) of Bi$_3$TiNbO$_9$. Note that there is a gradual variation in the relative intensity of the diffraction peaks at 14 and 24°, which are indexed as the (001) and (110) planes of Bi$_3$TiNbO$_9$, respectively. As analyzed in Supplementary Fig. 2b, the ratio of (001)/(110) shows a volcanic curve with the increase of W$^{6+}$ ion doping amount. Raman spectra were obtained with 532 nm wavelength excitation to detect the bonding situation and the structure of Bi$_3$TiNbO$_9$-*x*W (*x* = 0, 1%, 3%, 5%, 7%, 12%) nanosheets (Supplementary Fig. 3). The *E*$_g$ modes involving the stretching mode of diagonal oxygen ions on the *ab* plane of the BO$_6$ octahedron split into *B*$_{2g}$ at 533 cm$^{-1}$ and *B*$_{3g}$ at 573 cm$^{-1}$, which can be rationalized by the orthogonal distortion of Bi$_3$TiNbO$_9$[25–27]. The relative intensity of the two peaks increases with the W doping concentration in a certain range, which is in accord with the trend of ratio of (001)/(110) in XRD, pointing out that W$^{6+}$ ions are doped into the perovskite layers, thus contributing to structural distortion changes.

The element composition and chemical states of the Bi$_3$TiNbO$_9$-*x*W (*x* = 0, 5%) were determined by X-ray photoelectron spectroscopy (XPS) characterization. As shown in the Nb 3*d* spectrum of the Bi$_3$TiNbO$_9$-5%W (Fig. 2a), there are two additional peaks at 209.79 and 207.02 eV as compared with Bi$_3$TiNbO$_9$, of which the Nb 3*d* peak positions are in good agreement with those of Nb$_2$O$_5$ (Supplementary Fig. 4f). The chemical states of Bi 4*f*, Ti 2*p*, and O 1*s* of Bi$_3$TiNbO$_9$-W are nearly the same as those of Bi$_3$TiNbO$_9$ (Supplementary Fig. 4). These results reveal that W$^{6+}$ ions occupy the Nb$^{5+}$ ion sites in the perovskite structure, which is also supported by theoretical results in the following section of Fig. 3. Moreover, there are some impurity peaks that can be assigned to KNbWO$_6$ (PDF #25-0667) shown in the XRD diffraction pattern of Bi$_3$TiNbO$_9$-25%W with heavy doping (Supplementary Fig. 5), also affirming the substitution of Nb$^{5+}$ by W$^{6+}$. X-ray absorption spectroscopy (XAS) of the W *L*$_3$-edge was performed to probe the local coordination environment. The second derivative of X-ray absorption near-edge structure (XANES) spectrum in Supplementary Fig. 6 demonstrates two peaks with an energy difference of about 4.3 eV due to the split 5*d* states caused by the ligand field. And

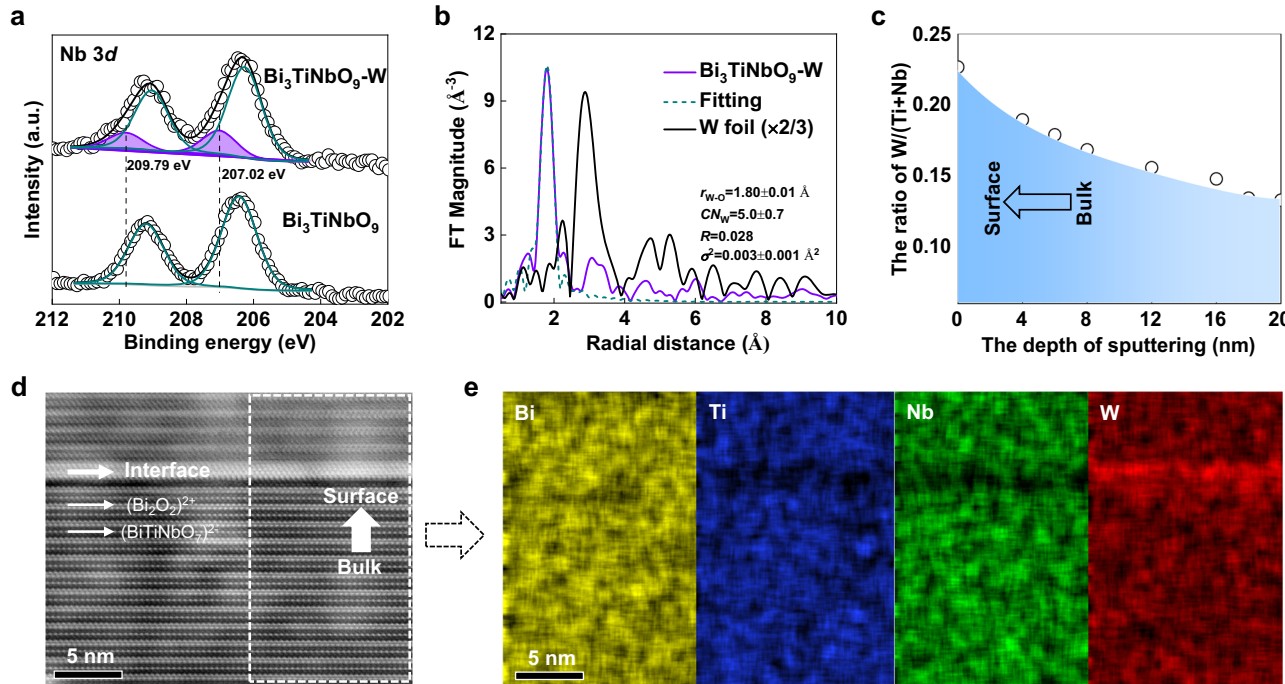

**Fig. 2 | The distribution of the doped element. a** Deconvolved XPS spectra of Nb 3*d*. **b** Fourier-transform curves of EXAFS data of W $L_3$-edge ($r_{W-O}$: The bond distance of W-O, $CN_W$: The coordination number of W, *R*: *R* factor, $\sigma^2$: Debye-Waller). **c** The ratio of W/(Ti+Nb) as a function of etching depth based on XPS of $Bi_3TiNbO_9$-W. **d** Atomic-level cross-section HAADF-STEM image of $Bi_3TiNbO_9$-W. **e** The corresponding EDS mapping images of elements (Bi, Ti, Nb, W).

the splitting gap is in agreement with the value of octahedral wolframates (3.0–5.6 eV)[28,29]. By further analyzing the Fourier transformed data of the extended fine structure region (EXAFS, Fig. 2b), the fitting curve reveals a W-O bond distance of $1.80 \pm 0.01$ Å and a coordination number of W atoms of $5.0 \pm 0.7$, which indicates the unsaturated coordination environment of W atoms in these $Bi_3TiNbO_9$-W nanosheets.

The distribution of W elements in $Bi_3TiNbO_9$-W nanosheets was confirmed by XPS depth profiles and elemental mapping analysis. As shown in Fig. 2c, the atomic ratio of W/(Ti+Nb) decreases gradually from 22.6% to 13.3% with increasing the etching depth by argon ions up to around 20 nm, proving the gradient doping of W. In addition, the combined XPS and ICP-OES investigations reveal that the W/(Ti+Nb) ratio on the surface is much higher than that in the bulk for all W-doped samples (Supplementary Fig. 7). Notably, the difference between the measured surface and bulk W/(Ti+Nb) ratio increases rapidly with W-doped concentrations of higher than 5%. These results suggest that the diffusion of doping atoms in the material is limited and the saturation doping concentration is about 3.5%. The spatial distribution of W was further examined by energy-dispersive X-ray spectroscopy (EDS). Viewing from Supplementary Fig. 8, the elements Bi, Ti, Nb, O, and W show a homogeneous distribution, demonstrating the homogenous doping of the W element in the basal {001} facet of the $Bi_3TiNbO_9$-W nanosheets. To further study the W distribution along the *c*-axis, the two stacked $Bi_3TiNbO_9$-W nanosheets were cut by focused ion beam, and their cross-section region was inspected by high-angle annular dark-field scanning transmission electron microscopy (HAADF-STEM, Supplementary Fig. 9). The layered structure of $Bi_3TiNbO_9$-W nanosheets can be clearly identified in Fig. 2d, in which the double-layer Bi atom is the $(Bi_2O_2)^{2+}$ layer and the single-layer Bi atom is the $(BiTiNbO_7)^{2-}$ layer, respectively. The microstructure at the interface between two nanosheets suggests that the outermost layer of $Bi_3TiNbO_9$-W is the $(Bi_2O_2)^{2+}$ layer. The corresponding EDS mapping images show a gradient W distribution feature in a single $Bi_3TiNbO_9$-W nanosheet,

with a maximal W content on the surface (Fig. 2e), further indicating the gradient doping of W along *c*-axis.

Density functional theory (DFT) calculations explored the effect of W doping on the electronic structure of $Bi_3TiNbO_9$. Based on the analysis of the ion valence and radius ($Ti^{4+}$: 0.605 Å, $Nb^{5+}$: 0.64 Å, $W^{6+}$: 0.60 Å), W atom may replace both the Ti and the Nb atoms in $Bi_3TiNbO_9$-W. Therefore, we have firstly calculated the formation energies of W-doped $Bi_3TiNbO_9$ ($2 \times 2 \times 1$) with one Ti or one Nb atom replaced by one W atom, aiming to identify the specific doping site for the W atom in $Bi_3TiNbO_9$. As a result, the formation energies are calculated to be 4.00 and 2.59 eV for W replacing Ti and Nb, respectively. This indicates an energetical preference for W dopants to occupy the Nb sites and is in good agreement with the XPS spectra (Fig. 2a) and XRD patterns (Supplementary Fig. 5). Therefore, we only consider cases of W replacing Nb for electronic structure calculations of $Bi_3TiNbO_9$-W. As shown in Fig. 3a, $Bi_3TiNbO_9$-W exhibits smaller bandgap values (2.35–2.50 eV) compared with that (2.79 eV) of pristine $Bi_3TiNbO_9$, due to a contribution from the W 5*d* orbitals with lower energy to the CBM (see Fig. 3b). Moreover, the bandgap values of $Bi_3TiNbO_9$-W decrease from 2.50 to 2.38, then to 2.35 eV with increasing the W-doping concentration from 3.2 to 6.3, then to 12.5%, respectively. UV-vis absorption spectra of $Bi_3TiNbO_9$-*x*W (*x* = 0, 1%, 3%, 5%, 7%, 12%) samples in Supplementary Fig. 10 also demonstrated a slight red shift of the absorption edge with the increase of W doping concentration.

As is well known, the core electrons of transition metals normally do not hybridize with the O 2*p* electrons in metal oxides, giving rise to the core electron level with very low energy. Such a core electron level does not obviously change by changing the chemical environment and has been widely adopted as a reference for band alignment[30,31]. Here, taking the Nb 4 *s* level as the reference, the band alignment of $Bi_3TiNbO_9$ and $Bi_3TiNbO_9$-W with different doping concentrations is given in Fig. 3b. As we can see, the valance band maximum (VBM) of $Bi_3TiNbO_9$-W exhibits a slight decrease with increasing the doping concentration. Combined with the gradually reduced bandgap values,

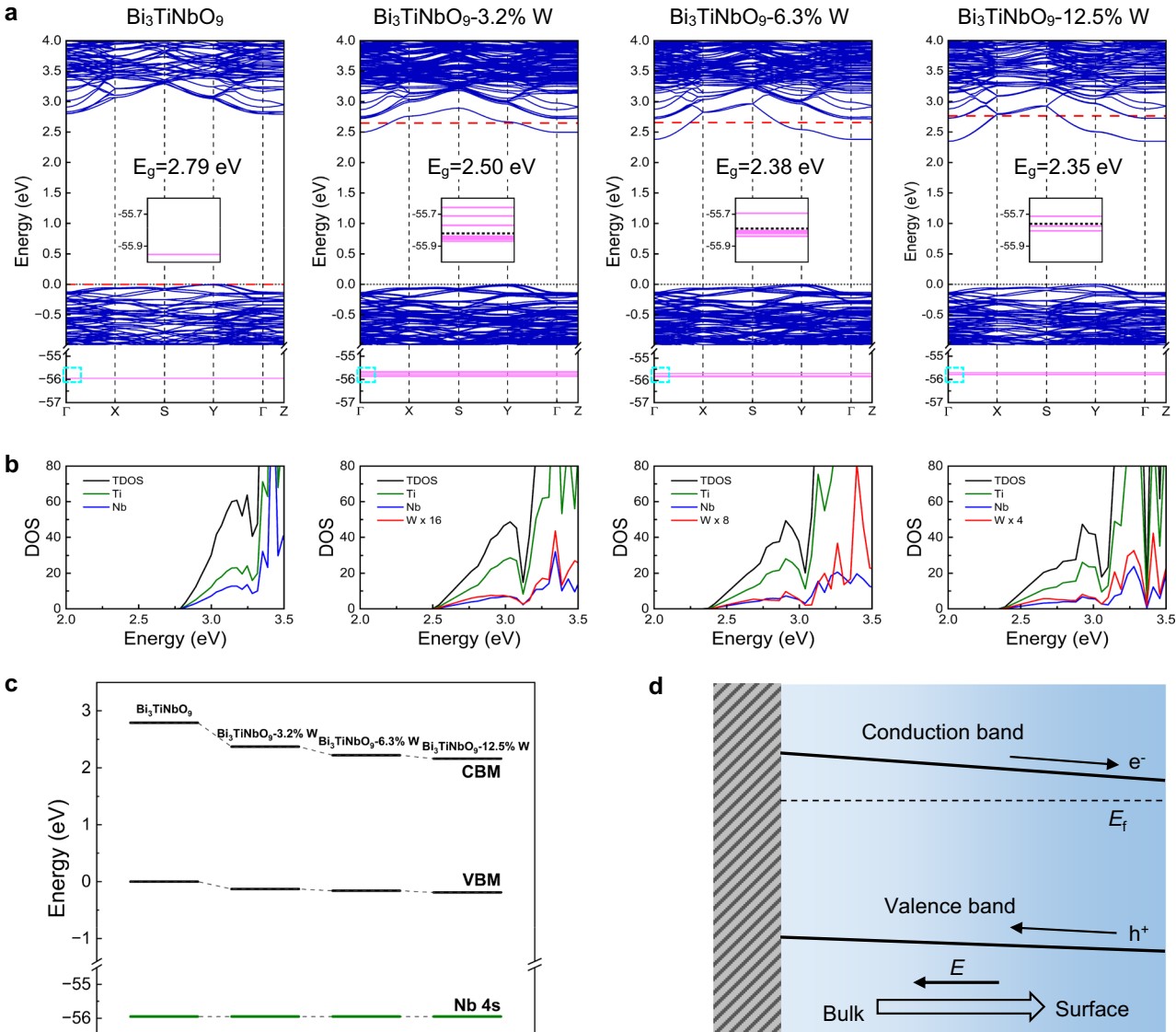

**Fig. 3 | Additional built-in electric field introduced by gradient doping. a** The calculated band structures of $Bi_3TiNbO_9$ with different W-doping concentrations. The calculated bandgap values are also given. The VBM (set to be 0 eV) and the Fermi level are denoted by black dotted and red dashed lines, respectively. The insets show an enlarged distribution of the Nb 4 s energy levels highlighted by the cyan dashed rectangles. The black dashed lines in the insets represent the averaged energy of the Nb 4 s levels. **b** The calculated density of state (DOS) of $Bi_3TiNbO_9$ with different W-doping concentrations. The DOS within an energy window from 2.0 to 3.5 eV nearby the CBM is shown in band structure plots. For the cases of B-doped $Bi_3TiNbO_9$ with doping concentration of 3.2, 6.3, and 12.5%, the local DOS value of W dopant is multiplied by a factor of 16, 8, and 4, respectively. **c** The band alignment of pristine $Bi_3TiNbO_9$ and W-doped $Bi_3TiNbO_9$ with different doping concentrations, by taking the Nb 4 s energy level as a reference. **d** Schematic diagram for the built-in electric field of $Bi_3TiNbO_9$ induced by gradient W-doping.

a more pronounced decrease of the conduction band minimum (CBM) can be observed with increasing the doping concentration for $Bi_3TiNbO_9$-W, compared with pristine $Bi_3TiNbO_9$ (Fig. 3c). Typically, the introduction of $W^{6+}$ ions in $Bi_3TiNbO_9$ could give rise to a shallow donor doping effect, caused by the unsaturated coordination environment of W atoms in these $Bi_3TiNbO_9$-W nanosheets as analyzed in Fig. 2b and can be confirmed by the Mott−Schottky curves shown in Supplementary Fig. 11. The gradient donor W doping could contribute to a gradual decrease in the gap between the Fermi level and the CBM from the bulk phase to the surface, forming an additional built-in electric field along the *c*-axis, which could provide a driving force for the migration of photoexcited electrons from bulk to the surface of $Bi_3TiNbO_9$-W (Fig. 3d)[32].

To understand the structural distortion caused by W doping, Rietveld refinement was carried out via the General Structure Analysis System (GSAS), as shown in Figs. 4a, b. There is a slight

shrinkage at the *a*-axis after W doping, while the *b*-axis maintains unchanged properties basically. The corresponding *b/a* value increases from 1.0055 to 1.0067, implying an enhancement of the polarization along the *a*-axis[33,34]. The piezoelectric coefficient ($d_{33}$) and dielectric constant ($\varepsilon_r$) were further used to investigate the ferroelectric polarization characteristics of the samples. Figures 4c, d shows the piezoelectric responses near the resonant frequency for $Bi_3TiNbO_9$ and $Bi_3TiNbO_9$-W samples at different driving voltages, respectively. It can be clearly observed that the piezoelectric resonant peak amplitude ($A_{max}$) increases with the driving voltages. The linear relationship between $A_{max}/Q$ and $V_{ac}$ can be obtained by fitting the harmonic oscillator model, where $Q$ is the quality factor and $V_{ac}$ is the driving ac voltage[35,36], as shown in Fig. 4e. The piezoelectric coefficient $d_{33}$ values are derived to be 1.24 and 2.29 pm/V for $Bi_3TiNbO_9$ and $Bi_3TiNbO_9$-W, respectively. The remarkable increase in $d_{33}$ indicates that the ferroelectric polarization intensity ($P_s$) of

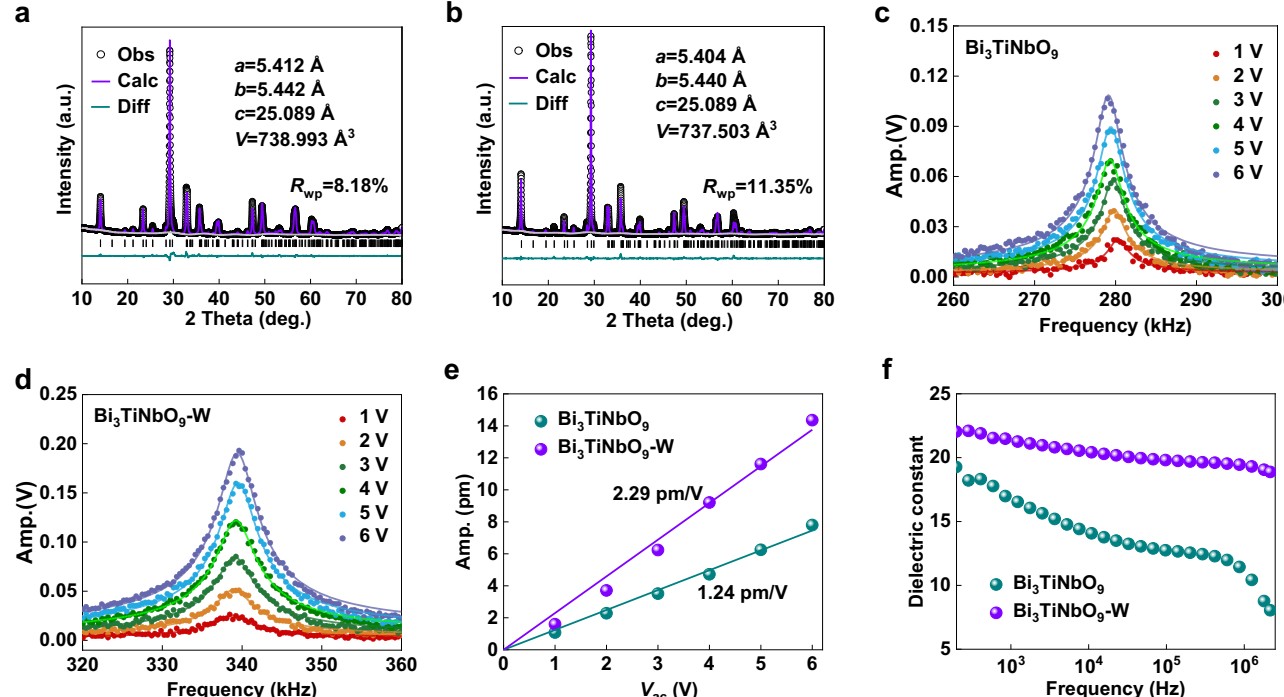

**Fig. 4 | Structural polarization characteristics.** X-ray diffraction Rietveld refinement: (**a**) $Bi_3TiNbO_9$, (**b**) $Bi_3TiNbO_9$-W. The piezoelectric response near the contact resonant frequency under different driving voltages: (**c**) $Bi_3TiNbO_9$, (**d**) $Bi_3TiNbO_9$-W. **e** The out-of-plane piezoelectric coefficient ($d_{33}$) values. **f** Dielectric constant versus frequency at room temperature of $Bi_3TiNbO_9$ (cyan point) and $Bi_3TiNbO_9$-W (purple point).

$Bi_3TiNbO_9$ is significantly enhanced after W doping[37,38]. Figure 4f shows the dielectric constant as a function of frequency at room temperature. Generally, the polarization of dielectric materials is contributed by electronic, ionic, orientation, and space charge polarization[39,40]. The contribution of space charge polarization decreases with the increase of frequency, giving rise to an inverse correlation with frequency for the dielectric constant (Fig. 4f). Comparing the dielectric constants of $Bi_3TiNbO_9$ and $Bi_3TiNbO_9$-W, it can be inferred that W doping could elicit an increase in the polarizability characteristics of $Bi_3TiNbO_9$, in accordance with the shrinkage of the $a$-axis by the Rietveld refinement (Fig. 4a, b) and the increased piezoelectric coefficient ($d_{33}$) values in Fig. 4e.

The difference of photogenerated charge transport behavior between $Bi_3TiNbO_9$ and $Bi_3TiNbO_9$-W was comprehensively analyzed. As shown in Fig. 5a, the electrons in $Bi_3TiNbO_9$ tend to migrate along the $c$-axis, but might be confined by interlayer barrier between the of $(Bi_2O_2)^{2+}$ and $(BiTiNbO_7)^{2-}$ layers, while the holes mostly migrate along the direction of the polarization electric field within the $ab$-plane[13,16]. The additional built-in electric field introduced by the gradient doping in $Bi_3TiNbO_9$-W could weaken the potential barrier between the layers of $(Bi_2O_2)^{2+}$ and $(BiTiNbO_7)^{2-}$, thus promoting the migration of the photogenerated electrons to the basal {001} surface. Meanwhile, the enhanced depolarization field of $Bi_3TiNbO_9$-W along the $a$-axis could improve the transport of the photogenerated holes to the lateral surface (Fig. 5b). The surface photovoltage (SPV) spectra of $Bi_3TiNbO_9$ and $Bi_3TiNbO_9$-W were executed to further explore the charge transfer characteristics (Fig. 5c). Owing to the restriction on the migration of the photogenerated electrons along $c$-axis, the $Bi_3TiNbO_9$ exhibits a positive SPV feature in a consistent optical response range (300–400 nm)[41–43]. In contrast, the negative signals of $Bi_3TiNbO_9$-W imply that the electron diffusion to the surface is largely promoted by the modified energy band structures, where the additional build-in electric field induced by the W gradient doping could facilitate the transport of the photogenerated electrons from bulk to the {001} surface. The difference in the charge transfer processes is responsible

for the higher hydrogen evolution activity and lower oxygen evolution activity of $Bi_3TiNbO_9$-W compared with that of $Bi_3TiNbO_9$ (Supplementary Fig. 12). The SPV signal on timescales of microseconds to milliseconds is attributed to the drift (Peak I) and diffusion (Peak II) processes of photogenerated carriers (Fig. 5d)[44]. $Bi_3TiNbO_9$ presents a positive peak II lasting for 12.3 ms before decaying to zero, while $Bi_3TiNbO_9$-W exhibits a negative peak II lasting for 16.7 ms. The extension of the photoresponse time might be ascribed to the enhancement of the polarization intensity along the $a$-axis direction and the additional electric field along the $c$-axis direction in $Bi_3TiNbO_9$-W, which can promote the anisotropic migration and effective separation of photogenerated charges.

The surface catalytic activity of $Bi_3TiNbO_9$ and $Bi_3TiNbO_9$-W, especially oxygen evolution reaction (OER), was also studied by DFT calculations. Considering its intrinsic ferroelectricity, there exists a depolarization field along the [$\bar{1}00$] direction of bulk $Bi_3TiNbO_9$ (see Supplementary Fig. 13a). In this case, the migration of photogenerated holes towards ($\bar{1}\bar{1}0$) and ($\bar{1}10$) facets can be accelerated by the depolarization field, making the ($\bar{1}\bar{1}0$) and ($\bar{1}10$) facets more active for water oxidation reaction due to the accumulation of holes[7,45]. Since the ($\bar{1}\bar{1}0$) facet and the ($\bar{1}10$) facet are equivalent in crystallography, only the reaction coordinate for OER on the ($\bar{1}\bar{1}0$) facet of $Bi_3TiNbO_9$ and $Bi_3TiNbO_9$-W was considered in this work (see Supplementary Fig. 13b). Supplementary Fig. 13c, d present the calculated free energy profiles of water oxidation on the ($\bar{1}\bar{1}0$) facets of $Bi_3TiNbO_9$ and $Bi_3TiNbO_9$-W. As we can see, $Bi_3TiNbO_9$ exhibits an overpotential of 0.78 V for OER at the Nb site, while the overpotential for OER at the W site on the ($\bar{1}\bar{1}0$) facet of $Bi_3TiNbO_9$-W is calculated to be 0.54 V. Basically, a lower overpotential indicates higher OER activity. However, the potential of photoexcited holes in $Bi_3TiNbO_9$ is roughly estimated to be -2.40 V[15], suggesting that the photoexcited holes can easily overcome the overpotentials for OER on the ($\bar{1}\bar{1}0$) facet of both $Bi_3TiNbO_9$ and $Bi_3TiNbO_9$-W. Moreover, as shown in Fig. 3d, the additional built-in electric field along $c$-axis induced by gradient W doping may impede the migration of photoexcited holes from bulk towards the surface as

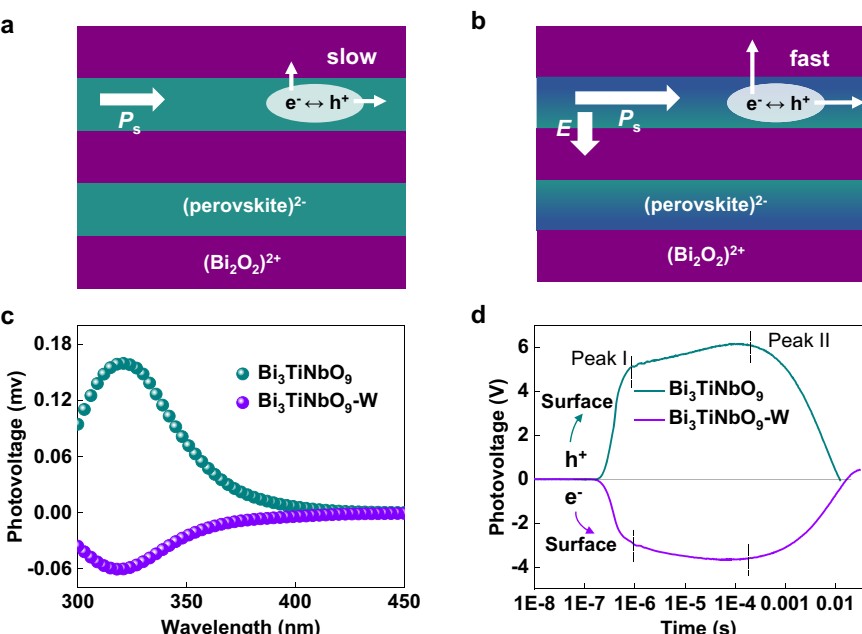

**Fig. 5 | Photogenerated charge transfer behavior.** Transfer of photogenerated charged carriers of $Bi_3TiNbO_9$ (**a**) and $Bi_3TiNbO_9$-W (**b**) along *ac* plane ($P_s$-ferroelectric polarization, *E*-additional electric field). Surface photovoltage (SPV) spectra (**c**) and transient photovoltage (TPV) response (**d**) of multiple $Bi_3TiNbO_9$ (cyan point) and $Bi_3TiNbO_9$-W (purple point) crystals.

validated by the SPV signal reversion from positive to negative after W doping (Fig. 5c, d). In fact, it has been reported that a low concentration of surface-reaching photogenerated holes can extremely limit the photocatalytic oxygen evolution performance[46]. Therefore, $Bi_3TiNbO_9$-W exhibits an even lower oxygen evolution yield (see Supplementary Fig. 12b) due to the compromise of the minor advantage of reduced OER overpotential and major disadvantage of less surface-reaching holes by W doping.

## Photocatalytic performance

Photocatalytic overall water splitting performance tests were carried out by loading $Rh/Cr_2O_3$ cocatalyst on the surface of $Bi_3TiNbO_9$ nanosheets[47–49]. As shown in Fig. 6a, the photocatalytic performance displays a volcanic curve with the W/(Ti+Nb) ratio increasing, which is consistent with the trend of structural distortion (inferred in Supplementary Fig. 2b, 3). When the ratio of W/(Ti+Nb) reaches 5%, that is, the saturation concentration of doping (Supplementary Fig. 7), $Bi_3TiNbO_9$-W exhibits the optimal photocatalytic overall water splitting activity with the average hydrogen and oxygen evolution rates of 43.99 and 20.66 $\mu mol\ h^{-1}$, respectively, which are 5.55 times higher than those of $Bi_3TiNbO_9$. Considering the very close specific surface area (1.2655 versus 1.3352 $m^2\ g^{-1}$) of the samples without and with W dopant, one can conclude that the additional built-in electric field has an essential effect on the photocatalytic overall water splitting activity enhancement. Once the doping amount exceeds the limit value, the formed impurity phase may be enriched on the surface of the sample, affecting light absorption characteristics and reducing photocatalytic activity. Furthermore, this gradient doping strategy in $Bi_3TiNbO_9$ system shows universality, demonstrating an efficient photocatalytic overall water splitting properties by employing Mo as the dopant as well. However, the activity of $Bi_3TiNbO_9$-Mo is only half of $Bi_3TiNbO_9$-W, as shown in Supplementary Fig. 14, which might be mainly attributed to the formation of defect transition levels below the conduction band edge induced by the Mo dopants. In contrast, W doping does not introduce any defect level in the band gap for the recombination of photogenerated carriers[50].

Time-dependent photocatalytic overall water splitting performance tests were used to evaluate the stability of $Bi_3TiNbO_9$-W. As shown in Figs. 6b, c, the activities of both $Bi_3TiNbO_9$ and $Bi_3TiNbO_9$-W decay rapidly during the long-term test, which can be ascribed to the self-reduction of $Bi^{3+}$ in the surface $(Bi_2O_2)^{2+}$ layers, as revealed in our previous work[15]. The XRD pattern shows that the diffraction peaks of $Bi_3TiNbO_9$ and $Bi_3TiNbO_9$-W remain unchanged after the photocatalytic overall water splitting reaction (Supplementary Fig. 15), indicating that the samples have a highly stable crystal structure. Adjusting the surface structure of $Bi_3TiNbO_9$ by acid etching and selectively exposing the robust perovskite layer (denoted as PL) have been successfully demonstrated to be an effective way to inhibit photocorrosion. We adopted such a strategy to remove the $(Bi_2O_2)^{2+}$ layer on the surface of $Bi_3TiNbO_9$-W (the resulting sample is named as $Bi_3TiNbO_9$-W-PL). The robust surface states of the obtained $Bi_3TiNbO_9$-W-PL catalysts can well balance the transfer of the photogenerated electrons to the perovskite layer-terminated surface and the photogenerated holes to lateral surface, corresponding to a weak and negative SPV signal demonstrated in Supplementary Fig. 16 Therefore, the $Bi_3TiNbO_9$-W-PL exhibits an efficient and stable activity (106.71 $\mu mol\ h^{-1}$ for $H_2$ and 47.94 $\mu mol\ h^{-1}$ for $O_2$) for overall water splitting in a 10 h long-term test (Fig. 6d), which is 10.4 times higher than that of pristine $Bi_3TiNbO_9$. The apparent quantum yield (AQY) of $Bi_3TiNbO_9$-W-PL was calculated to be 0.57% at 365 nm, being higher than the reported value for ferroelectric photocatalytic materials claimed (Supplementary Fig. 17, 18 and Supplementary Tables 1, 2).

## Discussion

In summary, we have successfully synthesized a series of W-doped $Bi_3TiNbO_9$ ($Bi_3TiNbO_9$-W) nanosheets. The concentration of this donor W dopant shows a gradient decrease from the surface to the bulk along the *c*-axis, which exhibits favorable characteristics for photocatalytic water splitting. One is that the gradient distribution of the W dopant can introduce an additional built-in electric field from the surface to the bulk to conquer the potential barrier between the layers of $(Bi_2O_2)^{2+}$ and $(BiTiNbO_7)^{2-}$ along the *c*-axis for the migration of the photogenerated electrons. The other is that ion

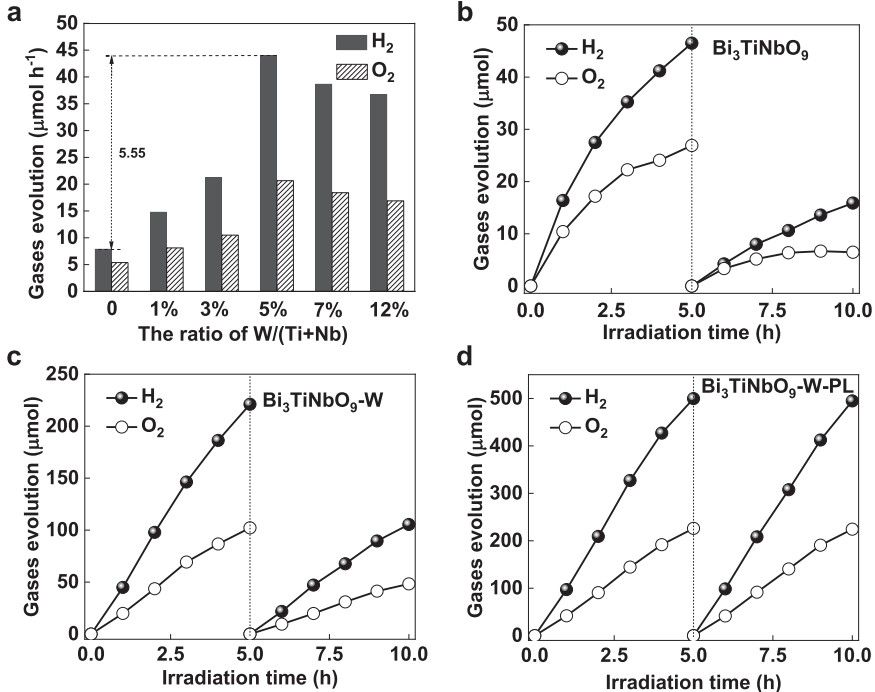

**Fig. 6 | Activity and stability of photocatalytic overall water splitting. a** Comparison of the activity of $Bi_3TiNbO_9$-W with different dopant concentrations. Stability test of photocatalytic overall water splitting ($\lambda \geq 300$ nm): (**b**) $Bi_3TiNbO_9$, (**c**) $Bi_3TiNbO_9$-W, (**d**) $Bi_3TiNbO_9$-W-PL.

substitution in the perovskite layer could strengthen the structural distortion of $Bi_3TiNbO_9$, which can enhance the depolarization field along the *a*-axis. As a result, the effective anisotropic migration and separation of the photogenerated carriers is fulfilled. Combined with the selective exposure of the robust $(BiTiNbO_7)^{2-}$ perovskite layer, our synthesized $Bi_3TiNbO_9$-W-PL catalyst shows an efficient and stable photocatalytic overall water splitting, which is 10.4 times higher than that of the pristine $Bi_3TiNbO_9$. This gradient doping strategy introduced in this work might provide guidance for the design of highly efficient and stable Aurivillius-type layered photocatalysts for overall water splitting.

# Methods
## Preparation of W-doped $Bi_3TiNbO_9$ nanosheets
$Bi_2O_3$ (0.699 g, Sinopharm (China)), anatase $TiO_2$ (0.08 g, Aldrich, 99%), $Nb_2O_5$ (0.133 g, Aladdin, 99.9%), NaCl-KCl (8 g, Sinopharm (China), a molar ratio of 1:1 and an eutectic temperature of 657 °C), and an appropriate amount of $WO_3$ (Sinopharm (China)) were mixed by an agate mortar for 30 min, and then transferred into an alumina crucible and calcined at 800 °C for 2 h. The obtained product was washed with hot deionized water to completely remove the flux reagents and dried in air, and denoted as $Bi_3TiNbO_9$-*x* W (*x* = 0, 1%, 3%, 5%, 7%, 12% in molar ratio). The synthesis process of $Bi_3TiNbO_9$-Mo is the same as that of $Bi_3TiNbO_9$-W, except that $MoO_3$ instead of $WO_3$ is used.

## Preparation of $Bi_3TiNbO_9$-W-PL nanosheets
250 mg of the $Bi_3TiNbO_9$-W sample was dispersed into 125 mL 0.32 mol $L^{-1}$ hydrochloric acid solution and continuously stirred for 5 h. The sample is then separated by filtration and washed with a large amount of deionized water until neutral.

## Cocatalyst deposition
150 mg of the sample was dispersed in 100 mL of aqueous methanol solution (10 vol%). Photoreduction of rhodium chloride (0.9 mg, $RhCl_3$, Alfa, 99.9%) and potassium chromate (0.9 mg, $K_2CrO_4$, Sigma-Aldrich,

99%) was carried out in steps under vacuum. The product was washed thoroughly with distilled water and dried.

## Sample characterization
The Scanning Electron Microscope (SEM, Nova NanoSEM 430) was used to examine the morphology of the samples. X-ray diffraction patterns of the samples were collected on a Rigaku D/max 2400 spectrometer using Cu *K*a X-rays of wavelength 1.54056 Å. The Raman spectra were collected on a Lab-Ram HR 800 spectrometer with an excitation wavelength of 532 nm. The UV-vis diffusion reflection spectra were obtained using a spectrophotometer (Jasco V-770) equipped with an integrating sphere in diffusion reflection mode. X-ray photoelectron spectroscopy (XPS) measurement was conducted on Escab-250 (Thermofisher Scientific, America) with a monochromatic Al *K*a X-ray source. All binding energies were referenced to the C 1 *s* peak (284.6 eV) that arises from adventitious carbon. XPS depth profile analyses were performed using an $Ar^+$ ion gun to etch stepwise layers at a rate of ~0.1 nm $s^{-1}$. Elemental analysis was carried out by a means of inductively coupled plasma atomic emission spectroscopy (ICP-OES). The piezoresponse force microscopy (PFM) of the powder samples were performed by a Scanning Probe Microscopy (SPM, Bruker Dimension Icon) system operated in Vertical PFM mode with Pt/Ir-coated Si cantilever tips. The dielectric measurements were performed using the two-probe impedance method with an Impedance Analyzer (TH2828A) over the frequency range from 100 Hz to 2 MHz. Cross-sectional HAADF-STEM images were acquired using an aberration-corrected STEM (Titan Themis 60–300 X-FEG microscope (FEI)) with double aberration (Cs) correctors from CEOS and a monochromator operating at 300 kV.

## Photocatalytic activity tests
Photocatalytic water splitting tests were conducted in an automatic testing system (Beijing Perfectlight Technology Co., Ltd., Labsolar-6A) with a 250 mL reaction container and a set temperature of 10 °C. The specific operation process is as follows: The sample loaded with cocatalyst (50 mg) was dispersed into 100 mL of pure water. After the

air in the system was completely removed, a 300 W xenon lamp (Beijing Perfectlight Technology Co., Ltd., PLS-SXE300D) at a wavelength of $\lambda \geq 300$ nm with an intensity of 349 mW cm$^{-2}$ was used as the light source. The gases generated from the system were analyzed at given time intervals by gas chromatography (Agilent Technologies, 6890 N).

## Surface Photovoltage (SPV) measurement

The system included a lock-in amplifier (SR830, Stanford Research Systems, Inc.) with a light chopper (SR540), a 500 W xenon lamp with a monochromator (SBP500, Zolix) as a light source, and a photovoltaic cell. The monochromatic light chopped with a frequency of 23 Hz is focused on the photovoltaic cell. The photovoltaic signal is amplified by a lock-in amplifier and transmitted to a computer for recording.

## Transient Photovoltage (TPV) measurement

The system included a Nd: YAG laser source (Q-smart 450 Quantel) providing laser radiation pulse (wavelength: 355 nm, pulse width: 5 ns and Intensity: 100 µJ), a 500 MHz digital phosphor oscilloscope (TDS 5054, Tektronix) with a preamplifier (5003 Brookdeal Electronics), and a parallel-plate capacitor-like sample chamber.

## Calculation detail

The density functional theory (DFT) calculations were performed by using Vienna Ab-initio Simulation Package (VASP)[51]. The interaction between ions and valence electrons was described using projector augmented wave (PAW) potentials[52], and the exchange-correlation interaction was described by generalized gradient approximation (GGA) with the Perdew-Burke-Ernzerhof (PBE) functional[53]. To overcome the well-known self-interaction error in DFT, the GGA + U method was used to treat the d electrons ($U = 4$eV for Ti 3$d$ and Nb 4$d$, $U = 3$eV for W 5$d$)[54]. The plane wave energy cutoff was set to be 500 eV. The valence electron configurations of Bi, Ti, Nb, O, and H atoms are $5d^{10}6s^26p^3$, $3s^23p^63d^24s^2$, $4s^24p^64d^45s^1$, $2s^22p^4$, and $1s^1$, respectively.

To describe the W-doped Bi$_3$TiNbO$_9$, a $2 \times 2 \times 1$ supercell containing 16 Ti and Nb atoms was constructed, subsequently, one, two, and four Nb atoms were replaced by W atoms, corresponding to a W-doping concentration of 3.2, 6.3, and 12.5%, respectively. In order to calculate the Gibbs free energy ($\Delta G$) of different intermediates (OH$^*$, O$^*$, OOH$^*$) adsorption on the ($\bar{1}\bar{1}0$) facet of Bi$_3$TiNbO$_9$, a simplified model only with the perovskite layer was adopted for the four-electron oxygen evolution reaction (OER), since the perovskite layer was demonstrated to be active for OER[55,56]. As for the W-doped case, one surface Nb atom was replaced by a W atom. For this model, except for one five-fold-coordinated Ti, Nb, or W atom on the ($\bar{1}\bar{1}0$) surface was left to be exposed as the reaction site for OER, all other unsaturated O and transition metal (Ti, Nb, W) atoms were saturated by H atoms and hydroxyl groups, respectively. The vacuum thickness was set to be 16 Å for both the $b$ and $c$ directions. The atomic structures were fully optimized with the convergence criteria for residual force and total energy of 0.02 eV Å$^{-1}$ and $10^{-6}$ eV, respectively.

The Gibbs free energy ($\Delta G$) was calculated from the DFT simulation by adding the free energy correction according to following equation,

$$\Delta G = \Delta E_{DFT} + \Delta G_{cor} \tag{1}$$

where the free energy correction ($\Delta G_{cor}$) for free molecules and adsorbed species was obtained by using the VASP toolkit[56], based on the following definition,

$$G_{cor} = ZPE + E_{vib} + E_{tran} + E_{rot} + PV - TS \tag{2}$$

$$G_{cor} = ZPE + E_{vib} - TS \tag{3}$$

where ZPE is the zero-point energy, $E_{vib}$, $E_{tran}$, and $E_{rot}$ are thermal corrections to the internal energy from vibrational, translational, and rotational motions of molecules, respective, the PV term converts internal energy to enthalpy, and the entropy term TS converts enthalpy to Gibbs free energy.

## Data availability

The data that support the findings of this study are available from the corresponding author upon request.

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

## Acknowledgements

This work was supported by the National Key R&D Program of China (no. 2021YFA1500800), National Natural Science Foundation of China (no. 51825204, 52120105003, 52188101), and CAS Projects for Young Sci-entists in Basic Research (YSBR-004), International Partnership Program of the Chinese Academy of Sciences (174321KYSB20200005). G. L. thanks the financial support from the New Cornerstone Science Foun-dation through the XPLORER PRIZE.

## Author contributions

G.L. led the project. Under the guide of G.L., J.H. with assistance of Y.Y. K. designed and performed the experiments. T.T.Y. and C.L.C. performed the HAADF-STEM tests. P.P.D. fitted EXAFS data. T.F.X. performed the

SPV and TPV tests. B.H.H and W.J.H. performed the measurements of piezoelectric coefficients and dielectric constants. Y.L. conducted FIB sample cutting. J.L. and L.C.Y. conducted theoretical calculations and wrote related contents. J.H. drafted most of the manuscript, and J.H.Q., Y.Y.K., L.C.Y. and G.L. edited the manuscript. All authors were involved in data analysis and manuscript revision.

## Competing interests

The authors declare no competing interests.

## Additional information

[1]Shenyang National Laboratory for Materials Science, Institute of Metal Research, Chinese Academy of Sciences, 72 Wenhua Road, Shenyang 110016, China. [2]School of Materials Science and Engineering, University of Science and Technology of China, 72 Wenhua Road, Shenyang 110016, China. [3]National Laboratory of Solid State Microstructures, College of Engineering and Applied Sciences, Nanjing University, Nanjing, Jiangsu 210093, China. [4]College of Chemistry, Jilin University, Changchun 130012, China. [5]Nanomaterials Centre, School of Chemical Engineering and Australian Institute for Bioengineering and Nanotechnology, The University of Queensland, St Lucia, QLD 4072, Australia. [6]Institute of Technology for Carbon Neutrality, Shenzhen Institute of Advanced Technology, Chinese Academy of Sciences, Shenzhen 518055, China. [7]These authors contributed equally: Jie Huang, Yuyang Kang, Jianan Liu. ✉e-mail: gangliu@imr.ac.cn

