## [Peer review file · Nature Communications]

REVIEWER COMMENTS

Reviewer #1 (Remarks to the Author):

The search for efficient and stable photocatalysts for overall water splitting has long been targeted and attracted worldwide attention in the past decades. Any important progress in this topic is highly expected and greatly welcome in solar photocatalysis community. This impressive study by Huang et al. reported a novel gradient tungsten-doped Bi₃TiNbO₉ ferroelectric photocatalyst that possesses not only the strengthened intrinsic depolarization field magnitude within the layers along a-axis but also an additional electric field along c-axis. This enables the desirable anisotropic migration of photogenerated electrons and holes towards the basal {001} surface and lateral {110} surface of the Bi₃TiNbO₉ nanosheets, respectively. As a result, the gradient doped ferroelectric photocatalyst with Rh/Cr₂O₃ cocatalyst shows an efficient and stable photocatalytic overall water splitting. Overall, the results achieved in this study are significant and could provide important implications for developing a new class of lead-free ferroelectric photocatalyst for overall water splitting. Moreover, the authors provide solid evidence for the clear relationship between structure (motivation) and activity (target) of photocatalysts by conducting comprehensive characterizations. Considering the high quality and potential great impact of this study, I am very glad to recommend its publication in Nature Communications. On the other hand, I also believe that the proper addressing of the following comments should be beneficial for this study.

- 1) For the substitutional doping of cations in metal oxides for metal dopants, the ionic radius and oxidation state of metal dopant are two most important parameters to be considered. The information on this concern is absent and needs to be added. It is also useful to discuss or predict the effect of other dopants in the same group of W dopant on the photocatalytic activity.
- 2) Substitutional tungsten doping in wide-bandgap metal oxide semiconductors usually causes some additional visible light absorption due to its low electron levels. According to UV-visible absorption spectra in Supplementary Figure 10, the W doped Bi₃TiNbO₉ also has additional marginal visible light absorption. Figure 5 shows the 5.5 times enhancement in photocatalytic overall water splitting under UV-visible light irradiation. How about the photocatalytic overall water splitting activity only under visible light irradiation? Please provide this information.
- 3) It is useful to compare the electronic structure of Bi₃TiNbO₉ before and after W doping. The authors are encouraged to conduct theoretical calculations and analyze the results. By doing this, the possible changes of both bandgap and charge mobility can be obtained.
- 4) Based on the detailed study of dopant concentration dependent photocatalytic overall water splitting activity (Fig. 5a), the optimal doping concentration of W to Ti and Nb was 5 at%. It is interesting to see that the maximum substitution of W to Ti(Nb) is also 5 at% in Supplementary Figure 7. This correlation should be further analyzed and discussed. This finding is very important for designing highly efficient doped photocatalysts.
- 5) It is remarkable that the surface photovoltage signal was reversed from positive to negative by the gradient W doping in Figure 4. This could be the strong evidence for the formation of an additional built-in electric field, which is the most important feature of the developed ferroelectric photocatalyst. The authors should highlight the correlation between the reversed photovoltage signal and formation of the additional built-in electric field.

Reviewer #2 (Remarks to the Author):

This manuscript prepared W doped Bi₃TiNbO₉ nanosheets by a modified flux method for photocatalytic overall water splitting. The preparation method and some characterizations are presented. Notably, a similar work has just been reported by the authors in Advanced Science (Selective Exposure of Robust Perovskite Layer of Aurivillius-Type Compounds for Stable Photocatalytic Overall Water Splitting, Adv. Sci. 2023, 2302206). In this work, the authors tried to dope a W-layer into the interlayer, which was not solidly supported the successful addition of this layer

with present results. The novelty of this work may be not high enough to be considered for publication on this journal.

Other comments are listed below,

1. The authors should highlight the innovation in introduction.
2. Whether the XRD peaks are shifted after W doping.
3. Add the scale in Fig. 2e and Supplementary Fig. 7
4. Please provide specific XPS etching conditions, and compare sample thickness with etching depth.
5. The author mentions that "the W/(Ti+Nb) ratio on the surface is much higher than that in the bulk for all W-doped samples". How does this affect the catalytic activity.
6. It is important to determine the stability of Ru single atomic sites during photocatalytic overall water splitting. It would be valuable to present high-resolution TEM images of W-doped Bi₃TiNbO₉ after photocatalytic water splitting to assess any changes in their structure.
7. The authors used the molten salt method to prepare the catalyst, what was the ratio of NaCl and KCl, what was the melting point, and what was the basis of the choice
8. What form is W doped in? Single atom? Could you elaborate further?
9. More specific experimental conditions need to be given, including light intensity, spectral range, reaction temperature, distance between lamp and reactor, etc.
10. The authors have done too little work on the water splitting mechanism, and should analyze the activity enhancement in depth with theoretical calculations
11. The authors should give a table to compare the performance of this work with other recently published works.
12. The data of the energy band structure of the materials should be supplemented to make the paper more convincing.
13. The manuscript will benefit from further careful proofreading to correct for errors in grammar, syntax and the selection of appropriate phrases to convey the intended message.

Reviewer #3 (Remarks to the Author):

Recommendation: Minor revisions

Comments:

This manuscript demonstrates a gradient doping strategy that introduces an additional electric field in the c-direction of the Bi₃TiNbO₉ nanosheets while simultaneously enhancing the polarized electric field in the a-direction. This strategy enhances the spatial separation efficiency of photogenerated carriers and achieves an efficient and durable overall water splitting. This manuscript provides a useful reference for the design of high-performance layered ferroelectric photocatalytic materials. I propose to publish it with minor revisions. The specific revisions are as follows:

1. In lines 75-76 of the manuscript, how to understand "modification" in the modified flux method? From the specific preparation process described by the authors, it is a conventional flux method.
2. Characterization of the ferroelectric polarization intensity (P_s) of single nanosheet is not easy, and what is the intrinsic physical mechanism of the piezoelectric coefficient d₃₃ characterizing the P_s of Bi₃TiNbO₉ nanosheets (Lines 171-173 of the manuscript)?
3. In lines 233-235 of the manuscript, the authors did not consider the effect of the specific surface area of the catalyst when comparing the hydrogen and oxygen evolution activities before and after doping. The effect of specific surface area can be excluded after using specific surface area normalization, and more accurate results can be obtained.
4. In lines 235-238 of the manuscript, the authors explain that once the doping exceeds the limit, the impurity phase formed may be enriched on the surface of the sample, thus affecting the light absorption properties and reducing the photocatalytic activity. However, from XRD and SEM, no impurity phase was observed even with doping up to 12%. In addition, the results of UV-visible absorption spectra showed that the higher the doping amount, the higher the light absorption (redshift). Therefore, are there some other reasons regarding the reduced catalytic activity due to overdoping?

Reply to Reviewers' Comments

Reviewer #1:

The search for efficient and stable photocatalysts for overall water splitting has long been targeted and attracted worldwide attention in the past decades. Any important progress in this topic is highly expected and greatly welcome in solar photocatalysis community. This impressive study by Huang et al. reported a novel gradient tungsten-doped $\text{Bi}_3\text{TiNbO}_9$ ferroelectric photocatalyst that possesses not only the strengthened intrinsic depolarization field magnitude within the layers along a-axis but also an additional electric field along c-axis. This enables the desirable anisotropic migration of photogenerated electrons and holes towards the basal $\{001\}$ surface and lateral $\{110\}$ surface of the $\text{Bi}_3\text{TiNbO}_9$ nanosheets, respectively. As a result, the gradient doped ferroelectric photocatalyst with Rh/ Cr_2O_3 cocatalyst shows an efficient and stable photocatalytic overall water splitting. Overall, the results achieved in this study are significant and could provide important implications for developing a new class of lead-free ferroelectric photocatalyst for overall water splitting. Moreover, the authors provide solid evidence for the clear relationship between structure (motivation) and activity (target) of photocatalysts by conducting comprehensive characterizations. Considering the high quality and potential great impact of this study, I am very glad to recommend its publication in Nature Communications. On the other hand, I also believe that the proper addressing of the following comments should be beneficial for this study.

Response: We thank the reviewer for the positive comments and constructive suggestions. Please find our point-to-point replies to your comments.

1. For the substitutional doping of cations in metal oxides for metal dopants, the ionic radius and oxidation state of metal dopant are two most important parameters to be considered. The information on this concern is absent and needs to be added. It is also useful to discuss or predict the effect of other dopants in the same group of W dopant on the photocatalytic activity.

Response: We thank the reviewer for this suggestion. The ionic radius and oxidation state of the dopant are important parameters affecting the doping sites. Following your suggestion, we have compared the relevant information in Supplementary Table 1. However, it is difficult to predict the effect of doped ions on photocatalytic activity, considering that the different local energy levels of dopants. Our XPS results and added theoretical results consistently suggest that W dopant tends to occupy Nb sites.

Supplementary Table 1 The cation radius of the perovskite layer in $\text{Bi}_3\text{TiNbO}_9$ and the VIB group in the periodic table of elements

ion species	Ti ⁴⁺	Nb ⁵⁺	Cr ³⁺	Mo ⁶⁺	W ⁶⁺
ionic radius (pm)	60.5	64	61.5	59	60

—Hosted by the Atomistic Simulation Group in the Materials Department of Imperial College

2. Substitutional tungsten doping in wide-bandgap metal oxide semiconductors usually causes some additional visible light absorption due to its low electron levels. According to UV-visible absorption spectra in Supplementary Fig. 10, the W doped $\text{Bi}_3\text{TiNbO}_9$ also has additional marginal visible light absorption. Fig. 5 shows the 5.5 times enhancement in photocatalytic overall water splitting under UV-visible light irradiation. How about the photocatalytic overall water splitting activity only under visible light irradiation? Please provide this information.

Response: The UV-visible absorption spectra (Supplementary Fig. 10) show that the absorption band edge position of $\text{Bi}_3\text{TiNbO}_9$ is 400.8 nm and the absorption band edge position of W doped $\text{Bi}_3\text{TiNbO}_9$ is about 414.8 nm. We tested the photocatalytic overall water splitting activity of W doped $\text{Bi}_3\text{TiNbO}_9$ under visible light irradiation ($\lambda > 400$ nm). However, no obvious activity was detected largely due to the weak visible light absorption of W doped $\text{Bi}_3\text{TiNbO}_9$.

3. It is useful to compare the electronic structure of $\text{Bi}_3\text{TiNbO}_9$ before and after W doping. The authors are encouraged to conduct theoretical calculations and analyze the results. By doing this, the possible changes of both bandgap and charge mobility can be obtained.

Response: Following this suggestion, we have added the DFT calculation of the electronic structure of $\text{Bi}_3\text{TiNbO}_9$ and $\text{Bi}_3\text{TiNbO}_9\text{-W}$ in revised Fig. 3. The results show that the bandgap values of W-doped $\text{Bi}_3\text{TiNbO}_9$ decrease with the increase of W-doping concentration, in which the valance band maximum (VBM) keeps nearly unchanged, and the conduction band minimum (CBM) is shifted downward. Consequently, a built-in electric field can be formed for $\text{Bi}_3\text{TiNbO}_9$ with gradient W doping. According to the spatial distribution of W-dopants achieved in this work, the built-in electric field induced by gradient W-doping is illustrated in revised Fig. 3d. Obviously, such an additional built-in electric field could provide a driving force for the migration of photoexcited electrons from bulk to the surface of W-doped $\text{Bi}_3\text{TiNbO}_9$.

Fig. 3 Additional built-in electric field introduced by gradient doping. **a**, The calculated band structures of $\text{Bi}_3\text{TiNbO}_9$ with different W-doping concentrations. The calculated bandgap values are also given. The VBM (set to be 0 eV) and the Fermi level are denoted by black dotted and red dashed lines, respectively. The insets show an enlarged distribution of the Nb-4s energy levels highlighted by the cyan dashed rectangles. The black dashed lines in the insets represent the averaged energy of the Nb-4s levels. **b**, The calculated density of state (DOS) of $\text{Bi}_3\text{TiNbO}_9$ with different W-doping concentrations. The DOS within an energy window from 2.0 to 3.5 eV nearby the CBM is shown in band structure plots. For the cases of B-doped $\text{Bi}_3\text{TiNbO}_9$ with doping concentration of 3.2, 6.3, and 12.5%, the local DOS value of W dopant is multiplied by a factor of 16, 8, and 4, respectively. **c**, The band alignment of pristine $\text{Bi}_3\text{TiNbO}_9$ and W-doped $\text{Bi}_3\text{TiNbO}_9$ with different doping concentrations, by taking the Nb-4s energy level as a reference. **d**, Schematic diagram for the built-in electric field of $\text{Bi}_3\text{TiNbO}_9$ induced by gradient W-doping.

4. Based on the detailed study of dopant concentration dependent photocatalytic overall water splitting activity (Fig. 5a), the optimal doping concentration of W to Ti and Nb was 5 at%. It is

interesting to see that the maximum substitution of W to Ti(Nb) is also 5 at% in Supplementary Fig. 7. This correlation should be further analyzed and discussed. This finding is very important for designing highly efficient doped photocatalysts.

Response: The improvement of photocatalytic overall water splitting activity of $\text{Bi}_3\text{TiNbO}_9$ caused by W^{6+} ion doping is mainly due to the enhanced built-in electric field promoting the transfer and separation of photogenerated carriers. Therefore, within the appropriate doping concentration range, the built-in electric field strength of the material is positively correlated with the doping concentration so that the photocatalytic activity is also continuously improved. Due to the limitation of structural tolerance, excessive dopants are difficult to be incorporated into the lattices and thus tend to be enriched on the surface of photocatalytic materials, limiting the photocatalytic activity. We also made relevant analysis in the manuscript. “As shown in Fig. 6a, the photocatalytic performance displays a volcanic curve with the $\text{W}/(\text{Ti}+\text{Nb})$ ratio increasing, which is consistent with the trend of structural distortion (inferred in Supplementary Fig. 2b, 3). When the ratio of $\text{W}/(\text{Ti}+\text{Nb})$ reaches 5%, that is, the saturation concentration of doping (Supplementary Fig. 7), $\text{Bi}_3\text{TiNbO}_9\text{-W}$ exhibits the optimal photocatalytic overall water splitting activity with the average hydrogen and oxygen evolution rates of 43.99 and 20.66 $\mu\text{mol h}^{-1}$, respectively, which are 5.55 times higher than those of $\text{Bi}_3\text{TiNbO}_9$. Once the doping amount exceeds the limit value, the formed impurity phase may be enriched on the surface of the sample, affecting light absorption characteristics and reducing photocatalytic activity.”

5. It is remarkable that the surface photovoltage signal was reversed from positive to negative by the gradient W doping in Figure 4. This could be the strong evidence for the formation of an additional built-in electric field, which is the most important feature of the developed ferroelectric photocatalyst. The authors should highlight the correlation between the reversed photovoltage signal and formation of the additional built-in electric field.

Response: Following this suggestion, we have added the following description in revised Supplementary Fig. 17. “Both $\text{Bi}_3\text{TiNbO}_9$ and $\text{Bi}_3\text{TiNbO}_9\text{-W}$ exhibit n type semiconductor characteristics with upward surface band bending. Since the interlayer barrier and the surface space charge layer restrict the migration of photogenerated electrons to the surface (equivalent to the inherent built-in electric field, E_i , pointing from the bulk phase to the surface), $\text{Bi}_3\text{TiNbO}_9$ presents a positive SPV signal. When the direction of the additional built-in electric field (E) induced by gradient doping is opposite to E_i , the positive SPV signal intensity will be weakened, and even the direction of the SPV response will be reversed, for example, the W gradient doping in $\text{Bi}_3\text{TiNbO}_9$ presents a negative SPV signal. However, when the direction of the additional

built-in electric field induced by gradient doping is the same as E_i , the positive SPV signal intensity will be enhanced”.

Reviewer #2:

This manuscript prepared W doped $\text{Bi}_3\text{TiNbO}_9$ nanosheets by a modified flux method for photocatalytic overall water splitting. The preparation method and some characterizations are presented. Notably, a similar work has just been reported by the authors in Advanced Science (Selective Exposure of Robust Perovskite Layer of Aurivillius-Type Compounds for Stable Photocatalytic Overall Water Splitting, Adv. Sci. 2023, 2302206). In this work, the authors tried to dope a W-layer into the interlayer, which was not solidly supported the successful addition of this layer with present results. The novelty of this work may be not high enough to be considered for publication on this journal.

Response: We greatly appreciate this reviewer for the great contribution in reviewing our manuscript, especially for the constructive comments and suggestions. Please find our point-to-point replies to your comments.

1. The authors should highlight the innovation in introduction.

Response: Following this constructive suggestion, we have highlighted the innovation of this study in the introduction of the revised manuscript as follows. “Although reducing the thickness of the layered material along the c-axis to obtain an ultrathin structure or selective exposure of different layers can shorten the migration distance of photogenerated electrons from the bulk to surface, the nature of poor interlayer charge transport remains unchanged. Therefore, the problem of large interlayer charge transport barrier remains a bottleneck in designing efficient layered $\text{Bi}_3\text{TiNbO}_9$ catalysts for overall water splitting. Here, we introduce an additional built-in electric field, perpendicular to the depolarization field in $\text{Bi}_3\text{TiNbO}_9$ nanosheets, by tungsten (W) element doping induced energy band structure adjustment between surface and bulk to break such a bottleneck. Typically, donor dopants can effectively increase the number of free electrons, thus raising the corresponding Fermi level of semiconductors.”

More importantly, the gradient W doping realized in this study causes two remarkable features that have never been realized in $\text{Bi}_3\text{TiNbO}_9$. One is the generation of an additional electric field along the c-axis and the other is the simultaneously enhanced magnitude of depolarization field within the layers along the a-axis due to strengthened structural distortion.

2. Whether the XRD peaks are shifted after W doping.

Response: After W ion doping, the diffraction peaks of $\text{Bi}_3\text{TiNbO}_9$ shift towards high angle in different degrees (Supplementary Fig. 2a). In order to quantitatively describe the change of crystal structure after doping, XRD patterns obtained at a slow sweep speed of $2^\circ/\text{min}$ (Fig. 3a and b) were collected for Rietveld refinement via the General Structure Analysis System. The results show that the crystal structure of $\text{Bi}_3\text{TiNbO}_9\text{-W}$ shrinks slightly along the a-axis compared with that of $\text{Bi}_3\text{TiNbO}_9$, which is consistent with the diffraction peak shift to higher angle in Supplementary Fig. 2a.

3. Add the scale in Fig. 2e and Supplementary Fig. 7

Response: Following your suggestion, we have added the scale bar in revised Fig. 2e and Supplementary Fig. 8.

4. Please provide specific XPS etching conditions and compare sample thickness with etching depth.

Response: Following your suggestion, we have added the following description about XPS etching conditions to the Methods section. However, considering the characteristics of the sample in powder, it is difficult to obtain the corresponding relationship between the sample thickness and the etching depth. By controlling the etching time of Ar^+ ions, the surface information of samples with different etching depths can be obtained.

5. The author mentions that “the $\text{W}/(\text{Ti}+\text{Nb})$ ratio on the surface is much higher than that in the bulk for all W-doped samples”. How does this affect the catalytic activity.

Response: Due to the limitation of structural tolerance and ion diffusion, the W ion concentration shows a gradient distribution that gradually decreases from the surface to the bulk. The resulting additional built-in electric field facilitates the transfer of photogenerated electrons across the interlayer, so it can improve the photocatalytic overall water splitting activity. In addition, in the appropriate doping concentration range, with the increase of W doping amount, the difference between the measured $\text{W}/(\text{Ti}+\text{Nb})$ of the surface and the bulk phase gradually increases (Supplementary Fig. 7), and the additional built-in electric field formed also increases. Therefore, the photocatalytic activity is positively correlated with the doping concentration. In addition, the effect of W doping on the surface catalytic activity of $\text{Bi}_3\text{TiNbO}_9$ was also studied by DFT calculations (revised Supplementary Fig. 13). The results show that W doping reduces the water oxidation overpotential of $\text{Bi}_3\text{TiNbO}_9$, indicating higher OER activity. However, the potential of photoexcited holes in $\text{Bi}_3\text{TiNbO}_9$ is roughly estimated to be ~ 2.40 V, suggesting that the photoexcited holes in $\text{Bi}_3\text{TiNbO}_9$ and $\text{Bi}_3\text{TiNbO}_9\text{-W}$ can easily overcome the overpotentials for

OER. Moreover, the additional built-in electric field induced by gradient W doping may impede the migration of photoexcited holes from bulk towards the surface (revised Fig. 3d). Therefore, the W-doped $\text{Bi}_3\text{TiNbO}_9$ exhibits an even lower oxygen evolution yield (Supplementary Fig. 11b).

6. It is important to determine the stability of Ru single atomic sites during photocatalytic overall water splitting. It would be valuable to present high-resolution TEM images of W-doped $\text{Bi}_3\text{TiNbO}_9$ after photocatalytic water splitting to assess any changes in their structure.

Response: Core-shell Rh/ Cr_2O_3 cocatalyst plays an important role in photocatalytic overall water splitting (*Angew. Chem. Int. Ed.* 2006, 118(46): 7970-7973). The primary particle size of the introduced Rh nanoparticles is 2–3 nm, which is coated with a Cr_2O_3 shell layer of about 2 nm thick. This typical cocatalyst structure is in various photocatalytic water splitting tests reported. The XRD pattern shows that the diffraction peaks of $\text{Bi}_3\text{TiNbO}_9$ and $\text{Bi}_3\text{TiNbO}_9$ -W remain unchanged after the photocatalytic overall water splitting reaction (revised Supplementary Fig. 16), indicating that the samples have a highly stable crystal structure.

7. The authors used the molten salt method to prepare the catalyst, what was the ratio of NaCl and KCl, what was the melting point, and what was the basis of the choice.

Response: The photocatalyst was prepared in NaCl-KCl eutectic molten salt. Its lowest melting temperature at the molar ratio of 1:1 was 657°C, which was significantly lower than the melting point of pure phases (776 °C for KCl and 801 °C for NaCl). In the process of material synthesis, the calcination temperature below or close to the melting point of molten salt can result in the insufficient ionic diffusion of precursors (*ACS Appl. Mater. Interfaces.* 2019, 11, 5642–5650). This poor crystal growth results in weaker photocatalytic activity. Although increasing the calcination temperature can improve the crystallinity of the synthesized material, the bismuth in the material is easy to evaporate at high temperatures, resulting in a higher concentration of defects. Therefore, NaCl-KCl eutectic molten salt with lower melting temperature is selected to synthesize the photocatalyst.

8. What form is W doped in? Single atom? Could you elaborate further?

Response: XRD patterns (Supplementary Fig. 5) and XPS spectra (Fig. 2a) show that W ions are incorporated into the lattice of $\text{Bi}_3\text{TiNbO}_9$ and occupy the Nb site, which has a BO_6 octahedral configuration. Usually, the doped element occupies the lattice site of the material in the form of a single atom. In addition, there is no W-W bond located at 2.85 Å in the EXAFS data of W L_3 -

edge of Bi₃TiNbO₉-W sample (Fig. 2b), indicating that the W dopant is incorporated into the lattice in the form of a single atom.

9. More specific experimental conditions need to be given, including light intensity, spectral range, reaction temperature, distance between lamp and reactor, etc.

Response: Following your suggestion, we have added the specific experimental conditions in the Methods section. The detailed parameters obtained are shown in revised Supplementary Table 2 and Figs.17 and 18. To show our results more clearly, we add the following description in the revised manuscript.

“Photocatalytic water splitting tests were conducted in an automatic testing system (Beijing Perfectlight Technology Co., Ltd., Labsolar-6A) with a 250 mL reaction container with a stationary temperature of 10 °C. The specific operation process is as follows: The sample loaded with cocatalyst (50 mg) was dispersed into 100 mL of pure water. After the residual air in the system was completely removed, a 300 W xenon lamp (Beijing Perfectlight Technology Co., Ltd., PLS-SXE300D) at a wavelength of $\lambda \geq 300$ nm with an intensity of 349 mW cm⁻² was used as the light source. The gases generated from the system were analyzed at given time intervals by gas chromatography (Agilent Technologies, 6890N).”

10. The authors have done too little work on the water splitting mechanism, and should analyze the activity enhancement in depth with theoretical calculations

Response: Following your suggestion, we have added the DFT calculations of water splitting in revised Supplementary Fig. 13. Considering that the hydrogen evolution reaction is carried out on the cocatalyst, the effect of W-doping on the oxygen evolution reaction activity was investigated. A smaller overpotential ($\eta = 0.54$ V) was observed at the W site of Bi₃TiNbO₉-W than that at the Nb site of Bi₃TiNbO₉ by 0.24 V, resulting in higher OER activity. However, the potential of photoexcited holes in Bi₃TiNbO₉ is roughly estimated to be ~2.40 V, suggesting that the photoexcited holes in both Bi₃TiNbO₉ and Bi₃TiNbO₉-W can easily overcome the overpotentials for OER. Moreover, the additional built-in electric field induced by gradient W doping may impede the migration of photoexcited holes from bulk towards the surface (revised Fig. 3d). Therefore, the W-doped Bi₃TiNbO₉ exhibits an even lower oxygen evolution yield for half reaction of water oxidation (Supplementary Fig. 12b). By comprehensive analysis, W-doped Bi₃TiNbO₉ showed superior photocatalytic overall water splitting activity than pristine Bi₃TiNbO₉, mainly due to the fact that the additional built-in electric field along the c-axis introduced by gradient doping can provide a driving force for the migration of photoexcited electrons from bulk to the surface (revised Fig. 5).

Supplementary Fig. 13 **a** Schematic diagram of the exposed $\{110\}$ facets of $\text{Bi}_3\text{TiNbO}_9$. The arrows indicate the spontaneous polarization (P_s) direction and the depolarization field (E) direction. **b** The atomic structures of simplified model for the $\{110\}$ facet of $\text{Bi}_3\text{TiNbO}_9$. From left to right, adsorption configurations for three intermediates (OH^* , O^* , OOH^*) of the four-electron oxygen evolution reaction on the $\{110\}$ facet of $\text{Bi}_3\text{TiNbO}_9$ are presented. The purple, green, blue, light red, and white balls represent Bi, Ti, Nb, O, and H atoms, respectively. The O and H atoms of intermediates highlighted in the dotted ellipses are denoted by bright red and yellow balls, respectively. The DFT calculation of the reaction coordinate of the four-electron oxygen evolution reaction at different active sites (Ti, Nb, W) on the facet $\{110\}$: **c** $\text{Bi}_3\text{TiNbO}_9$, **d** $\text{Bi}_3\text{TiNbO}_9\text{-W}$.

11. The authors should give a table to compare the performance of this work with other recently published works.

Response: We thank the reviewer for this suggestion. Following your suggestion, we have added the following performance comparison in Supplementary Table 3. Ferroelectric materials (e.g., PbTiO_3 , BiFeO_3 and BaTiO_3) with built-in electric field that could facilitate photogenerated carrier separation have attracted wide attention in the field of photocatalysis. Aurivillius compounds with layered structure, such as $\text{Bi}_3\text{TiNbO}_9$, have great potential in photocatalytic overall water splitting due to their easily modified crystal structure and tunable ferroelectric

properties. Photocatalytic overall water splitting in this work achieved an apparent quantum yield (AQY) of 0.57% at 365 nm, which is higher than the reported value for ferroelectric materials claimed.

Supplementary Table 3 Performance comparison of W-doped Bi₃TiNbO₉ with previously reported ferroelectric photocatalysts.

Photocatalysts	H ₂ evolution rate ($\mu\text{mol h}^{-1}$)	O ₂ evolution rate ($\mu\text{mol h}^{-1}$)	AQE (%)	Ref
Bi ₃ TiNbO ₉ -W	106.71	47.94	0.57% (365 nm)	This work
Bi ₃ TiNbO ₉	21.78	9.94	0.26% (365 nm)	2
PbTiO ₃	3.29	1.74	0.071 (365 nm)	3
BaTiO ₃ /Au	$\sim 0.89 \mu\text{mol cm}^{-2}$	$\sim 0.47 \mu\text{mol cm}^{-2}$	—	4
Bi ₃ TiNbO ₉	17.13	13.76	—	5*
Bi ₃ Ti _{0.8} Cr _{0.1} Nb _{1.1} O ₉	41.11	—	0.52% ($\geq 250 \text{ nm}$)	1*
Bi ₃ TiNbO ₉ /RGO	2.4	—	—	6*

* represents the photocatalytic water splitting half reaction

12. The data of the energy band structure of the materials should be supplemented to make the paper more convincing.

Response: Following your suggestion, we have added the DFT calculations of energy band structure of the materials in revised Fig. 3. The results show that the bandgap values of W-doped Bi₃TiNbO₉ decrease with the increase of W-doping concentration, in which the valance band maximum (VBM) remains nearly unchanged and the conduction band minimum (CBM) shifts downward. Consequently, a built-in electric field can be expected for W-doped Bi₃TiNbO₉ with different doping concentrations. According to the spatial distribution of W-dopants achieved in this work, the built-in electric field induced by W-doping can be illustrated in revised Fig. 3d. Obviously, such a built-in electric field may provide a driving force for the migration of photoexcited electrons from bulk to the surface of W-doped Bi₃TiNbO₉.

Fig. 3 Additional built-in electric field introduced by gradient doping. **a**, The calculated band structures of $\text{Bi}_3\text{TiNbO}_9$ with different W-doping concentrations. The calculated bandgap values are also given. The VBM (set to be 0 eV) and the Fermi level are denoted by black dotted and red dashed lines, respectively. The insets show an enlarged distribution of the Nb-4s energy levels highlighted by the cyan dashed rectangles. The black dashed lines in the insets represent the averaged energy of the Nb-4s levels. **b**, The calculated density of state (DOS) of $\text{Bi}_3\text{TiNbO}_9$ with different W-doping concentrations. The DOS within an energy window from 2.0 to 3.5 eV nearby the CBM is shown in band structure plots. For the cases of B-doped $\text{Bi}_3\text{TiNbO}_9$ with doping concentration of 3.2, 6.3, and 12.5%, the local DOS value of W dopant is multiplied by a factor of 16, 8, and 4, respectively. **c**, The band alignment of pristine $\text{Bi}_3\text{TiNbO}_9$ and W-doped $\text{Bi}_3\text{TiNbO}_9$ with different doping concentrations, by taking the Nb-4s energy level as a reference. **d**, Schematic diagram for the built-in electric field of $\text{Bi}_3\text{TiNbO}_9$ induced by gradient W-doping.

13. The manuscript will benefit from further careful proofreading to correct for errors in grammar, syntax and the selection of appropriate phrases to convey the intended message.

Response: We thank the reviewer for this suggestion. We have carefully reviewed the manuscript and corrected the mistake in the revised manuscript.

Reviewer #3:

This manuscript demonstrates a gradient doping strategy that introduces an additional electric field in the c-direction of the Bi₃TiNbO₉ nanosheets while simultaneously enhancing the polarized electric field in the a-direction. This strategy enhances the spatial separation efficiency of photogenerated carriers and achieves an efficient and durable overall water splitting. This manuscript provides a useful reference for the design of high-performance layered ferroelectric photocatalytic materials. I propose to publish it with minor revisions. The specific revisions are as follows:

Response: We thank the reviewer for the positive comments and constructive suggestions. Please find our point-to-point replies to your comments below.

1. In lines 75-76 of the manuscript, how to understand “modification” in the modified flux method? From the specific preparation process described by the authors, it is a conventional flux method.

Response: We are grateful to the referee for pointing out the inappropriate description here and revised relevant description in the manuscript.

2. Characterization of the ferroelectric polarization intensity (P_s) of single nanosheet is not easy, and what is the intrinsic physical mechanism of the piezoelectric coefficient d_{33} characterizing the P_s of Bi₃TiNbO₉ nanosheets (Lines 171-173 of the manuscript)?

Response: Duran *et al.* proposed that the piezoelectric properties of a ferroelectric ceramic can be expressed as, $d_{33} \sim 2Q\epsilon_0\epsilon P_s$, where Q is the electrostrictive coefficient, ϵ_0 is the permittivity of free space, ϵ is the dielectric constant, and P_s is the remanent polarization (*J. Electroceramics*, 2003, 10, 47-55). Therefore, the piezoelectric coefficient (d_{33}) can be used to reflect the polarization intensity (P_s) of Bi₃TiNbO₉ nanosheets. In addition, the X-ray diffraction Rietveld refinement also shows that the crystal distortion of Bi₃TiNbO₉-W along the a-axis direction is enhanced compared with Bi₃TiNbO₉, which is consistent with the change trend of the piezoelectric coefficient (d_{33}) results (revised Fig. 4).

3. In lines 233-235 of the manuscript, the authors did not consider the effect of the specific surface area of the catalyst when comparing the hydrogen and oxygen evolution activities before

and after doping. The effect of specific surface area can be excluded after using specific surface area normalization, and more accurate results can be obtained.

Response: Following your suggestion, we have measured specific surface area of the samples before and after W doping. Their very close surface area (1.2655 versus 1.3352 m² g⁻¹) of undoped and doped samples can safely rule out the effects of specific surface area on photocatalytic activity.

4. In lines 235-238 of the manuscript, the authors explain that once the doping exceeds the limit, the impurity phase formed may be enriched on the surface of the sample, thus affecting the light absorption properties and reducing the photocatalytic activity. However, from XRD and SEM, no impurity phase was observed even with doping up to 12%. In addition, the results of UV–visible absorption spectra showed that the higher the doping amount, the higher the light absorption (redshift). Therefore, are there some other reasons regarding the reduced catalytic activity due to overdoping?

Response: When the theoretical W doping amount is higher than 5 %, the actual measured surface W content increases significantly (Supplementary Fig. 7), indicating that the excessive W element is difficult to continue to be incorporated into the Bi₃TiNbO₉ lattice. And the appropriate W doping basically does not affect the light absorption range of Bi₃TiNbO₉ (Supplementary Fig. 10). When the doping amount is far from the saturation value, the absorption edge position is obviously red-shifted, indicating that the W-containing related impurity phase with wide spectral response and non-photocatalytic activity exists on the surface of Bi₃TiNbO₉, which hinders the light absorption of the photocatalyst and thus reduces the photocatalytic activity. It is difficult to observe the impurity phase by XRD and SEM mainly because the content of the impurity phase is lower than the detection limit of the instrument. When the theoretical doping amount reaches 25 %, the KNbWO₆ impurity phase is obviously observed (Supplementary Fig. 5).

REVIEWERS' COMMENTS

Reviewer #1 (Remarks to the Author):

The authors have addressed my concerns satisfactorily. I think this manuscript can be published as it is.

Reviewer #2 (Remarks to the Author):

The authors revised the manuscript with all previous comments addressed proeprly. I suggest to accept it for publication on this journal.

Reviewer #3 (Remarks to the Author):

Comments :

The authors have carefully revised the manuscript based on my comments and those of the other reviewers, and I believe this work is now ready for publication.